# Barriers to access and unmet needs in mental health care for Venezuelan migrants in a southern border region of Colombia: The experiences of community workers

John Fitton[1]¤*, Julie Benavides-Melo[2], Christine Greenhalgh[1]

**1** Division of Population Health, Health Services Research and Primary Care, School of Health Sciences, Faculty of Biology, Medicine and Health, The University of Manchester, Manchester, United Kingdom, **2** Interdisciplinary Research Group in Health-Disease (GIISE), Faculty of Medicine, Universidad Cooperativa de Colombia, Pasto, Colombia

¤ Current address: Institute for Global Health, University College London, London, United Kingdom
* john.fitton.24@ucl.ac.uk

## Abstract

This study investigates the experiences of community workers who support Venezuelan migrants to access mental health care in Nariño, Colombia, an important migratory site on the southern border with Ecuador. The Colombian state has done much to enable healthcare access for the 2.86 million Venezuelan migrants in the country, but major barriers to services remain. This qualitative study explores access to services and unmet needs in mental health for migrants in this region. A phenomenological approach was adopted. Participants were selected from a range of frontline community organisations. Semi-structured interviews were completed remotely and reflexive thematic analysis of the data was performed. The timing of this study - during a period of reduction of funding for humanitarian projects in the region – permits insights into the concerns of community workers facing a critical gap in mental health care for migrants with irregular status, particularly a striking lack of services for unaccompanied men in transit. This study identifies concerns about the pervasive nature of psychoactive substance use in unaccompanied men in transit, raising an urgent need for further research. We found that competing socioeconomic priorities prevent migrants from attending to their mental health and additionally reflect on specific challenges associated with the condition of transit in this border region. This study supports previous research, which has noted lack of knowledge by health providers of the legal rights of migrants to healthcare, and adds complexity by exploring conflicting viewpoints on allegations of institutional discrimination.

**Data availability statement:** The data for this qualitative study consist of the transcripts of participant interviews (pseudonymised). Excerpts of the data relevant to the research findings are included in the body of the article. Ethically, it is not possible to make the full transcripts publicly available, as even despite pseudonymisation it might be possible to identify participants from the full transcripts. Legally, participants were informed as part of the informed consent process that the pseudonymised transcripts would be held securely rather than published. Patient data (including transcripts) are held in the research drive of the University of Manchester. Requests to access the data should be directed to the corresponding author (Dr John Fitton), or to Dr Christine Greenhalgh (Christine.greenhalgh@manchester.ac.uk) or to the Information Governance office of the University of Manchester (dataprotection@manchester.ac.uk).

**Funding:** The authors received no specific funding for this work.

**Competing interests:** The authors have declared that no competing interests exist.

## 1. Introduction

Colombia hosts 2.86 million of the 7.77 million Venezuelans who have left their country since 2015 due to its economic, political and humanitarian crisis [1]. Venezuelan migrants are known to experience high rates of mental health problems including depression, anxiety and posttraumatic stress [2], exacerbated by sociocultural challenges and the experience of discriminatory and xenophobic behaviours [3]. Colombia has been described as a "global leader" in its response to Venezuelan migration, having "set new standards for integration of large flows of people in conditions of forced displacement and heightened vulnerability" into the health and social support system [4]. Despite this achievement, there remain major challenges for Venezuelans who attempt to access the Colombian healthcare system [5–10]. This qualitative study investigates access to care and unmet needs in mental health for Venezuelan migrants in Colombia's southern border region of Nariño, from the perspective of people who work to support migrants in local community organisations.

## 2. Background

Migration is a key determinant of mental health and immigrants may experience more depression, anxiety and somatic disorders than native populations [11]. However, studies of the mental health of migrants in low- and middle-income countries (LMICs) have been noted as "severely lacking", representing a significant research gap [12]. In Colombia, a national household survey of Venezuelans in 2023 found that 82% had symptoms associated with poor mental health (principally fear, anxiety and depression) [13]. Studies using clinical screening tools for mental health problems have reported high rates of significant depression in Venezuelans in various settings in Colombia [14–16]. Qualitative studies of general health perceptions of Venezuelans in Colombia have reported evidence of depression, anxiety and stress, particularly in relation to the process of migration and "migratory grief" [7,17].

Colombia's healthcare system is based on a national insurance scheme: Colombian citizens are affiliated with health insurance agencies that commission health services (primary and specialist) from provider institutions [18]. Patients may access mental health care by attending a primary care physician who can refer to specialist services, by attending the emergency department directly without referral, or by paying for a private service. Delivery of mental health services for Colombian citizens shows marked inequities, with particularly stark geographical disparities between urban and rural areas [19].

Colombia has introduced numerous policies to improve access of Venezuelan migrants to healthcare, with arguably the most important being the Temporary Protection Status for Venezuelan Migrants (TPSV) in March 2021 [4]. This provided an accelerated regularisation of migratory status for Venezuelans, with a permit to stay for up to ten years, and provision of the Temporary Protection Permit (TPP) which allows affiliation with healthcare insurers and access to the full range of health services in the same way as Colombians. However, the TPSV's restrictions include that it is not available to migrants who entered via informal routes after January 2021 or to any migrants who entered after May 2023 (with the exception of children registered

in the educational system) [20]. Government data from December 2023 show that of the total 2.86 million Venezuelans in Colombia, 1.36 million had achieved affiliation with the health system [21].

Venezuelans who are unable to affiliate with the health system – including all migrants with irregular status - have access to state-provided mental health care only via the emergency department. For this population, non-governmental humanitarian organisations – international and national – have played a crucial role in the provision of healthcare, including psychological care [6].

In addition to irregular legal status, key barriers to healthcare for Venezuelan migrants in Colombia which have been identified in previous research include limited awareness of the legal entitlement to healthcare and how to navigate the system by migrants themselves [5,8,22], knowledge gaps among staff in healthcare institutions [6,10], xenophobic discrimination in healthcare institutions [8,10,23] and lack of help-seeking by migrants due to competing socioeconomic priorities [7,8,24].

This study aims to investigate the challenges experienced by community workers when supporting Venezuelan migrants to access care for mental health problems in Nariño, Colombia. We follow the conceptualisation of healthcare access formulated by Levesque et al [25] as the opportunity to identify healthcare needs, to seek healthcare, to reach or use services, and to have a need for services fulfilled. This approach incorporates both service accessibility characteristics (approachability, acceptability, availability, affordability and appropriateness) and corresponding population abilities (to perceive, seek, reach, pay and engage). Barriers to access are understood here as the individual, institutional or systemic factors which prevent migrants from accessing services to which they are legally entitled. Additionally, by exploring "unmet needs", the study investigates mental health needs identified by community workers for which services are not available.

## 3. Study setting

This study is located in the Colombian department of Nariño, which is a key site on the migratory route as it lies both on the southern border with Ecuador and on the Pacific coastline. In December 2023, there were 43.5 thousand Venezuelans in Nariño, with the majority in the cities of Ipiales and Pasto [26]. Prior to 2023, the main flow of Venezuelan migrants was moving southward across the border to leave Colombia; however, during 2023, there was a marked increase in Venezuelans crossing the border northward into Colombia with the intention to return to Venezuela or to travel towards Central and ultimately North America [26]. Precise numbers of migrants in transit through Nariño are difficult to obtain because it is estimated that 85% have entered via illegal border crossings [27]. However, approximately 46% of irregular migrants in transit in Colombia have entered from Ecuador and, therefore, are likely to have passed through Nariño [28]. The majority of migrants in transit in Nariño are travelling on foot (47%) or in cargo transportation (41%), with just 9% using public transport [29]. Non-governmental humanitarian care for Venezuelan migrants in Nariño has been provided by organisations under the coordination of the international Inter-agency Group on Mixed Migration Flows (Grupo Interagencial Sobre Flujos Migratorios Mixtos) and has included hostels, food kitchens, and support shelters on the transit route [26].

Identification of the research question was the result of a series of meetings held by JF and JBM with stakeholders in Nariño – including people working in charities, non-governmental organisations (NGOs) and government agencies – in July 2023 and February 2024. During this work, we recognised that there is much to be learned from the frontline community workers who have supported very large numbers of Venezuelan migrants in this region. We identified that they offer important insights into the challenges of working with migrants, including navigating the healthcare system, ethical dilemmas and coping with the effects of policy change.

We used a broad definition of "community workers" in order to access a varied range of experiences in what is a relatively small workforce in Nariño. For the purposes of this study, "community workers" are people who work in the frontline with Venezuelan migrants at community level. We do not use the term to refer to a specific role or profession, but rather as a general term which can include workers in international agencies, NGOs and government employees. The community workers may be of any nationality, including Venezuelan (given that many NGOs in Colombia are run by resident

Venezuelans). Their work is of various types but typically consists of screening for urgent health and social needs (for example, at transit points or in hostels) and outreach activities aimed to help migrants navigate available health, financial and social resources. We sought participants working both with migrants in residence and in transit, because in reality in Nariño the division between these types of migrant status is often fluid. The roles of our final sample are described in 5.1.

We would additionally have liked to interview Venezuelan migrants directly about their own experiences of attempting to access mental health care. However, we concluded this was not possible in the current study due to concerns both practical (the challenge of arranging in-depth interviews with a population in rapid transit) and ethical (the risk of causing harm through re-traumatisation in a highly vulnerable population with scarce opportunity for follow-up support).

## 4. Methodology

### 4.1 Ethics statement

Ethical approval was obtained on 9 January 2024 from the Proportionate University Research Ethics Committee of the University of Manchester (Ref: 2024-18935-32506), including approval of the consent procedure (described in 4.2).

This study adopted a phenomenological approach, aiming to understand the world as it is experienced by the community workers, through their personal perspectives. Qualitative research rooted in phenomenology can explore, describe and interpret how the participants make sense of their own lived experience while questioning pre-conceived assumptions [30]. As well as describing their own experiences, the participants were frequently also keen to narrate second-hand accounts of the experiences of migrants. Where these migrant accounts are included in the analysis, the focus in the study is on their impact and implications for the community workers, viewed through the lens of their lived experience.

The research formed part of JF's Master of Public Health degree from the University of Manchester. JBM, from the Universidad Cooperativa de Colombia, Pasto, was a co-supervisor of the project. The study findings are reported with acknowledgement of the COREQ checklist [31], but we have preferentially followed Braun and Clarke's Reflexive Thematic Analysis Reporting Guidelines (RTARG) [32] for greater methodological coherence.

### 4.2 Participant selection and recruitment

The inclusion criteria for participants were: age over 18 years; working (either in a paid or voluntary capacity) in Nariño in an international agency, NGO or government agency; working in a role which directly supports Venezuelan migrants to identify their health needs or to navigate health, financial and social resources; and with a stable internet connection in order to complete an online interview.

Potential participants were excluded if they were healthcare clinicians (doctors, nurses and psychologists), since our aim was to explore non-specialist views of mental health challenges. (We did not exclude non-clinicians who had undergone mental health training as part of their job, as such training is very common in frontline roles.) We did not recruit employees in healthcare institutions, as our research question addressed barriers to healthcare access and unmet needs rather than the experiences of those who receive care. We also excluded people who worked in a managerial or supervisory role in an organisation, without regular direct contact with migrants.

We purposively selected participants to provide viewpoints rich in information, with experience of working with migrants both in transit and in residence, and from a range of organisation types including Venezuelan-led. Recruitment was supported by gatekeepers who included local government officers, representatives of humanitarian agencies and Venezuelan community leaders. Recruitment took place from 19 February 2024 to 18 May 2024. Potential participants were sent an introductory email (including a Participant Information Sheet) and were offered a brief explanatory meeting prior to providing informed written consent. All participants were adults participating in their professional capacity, therefore formal assessment of capacity to consent was not required. Some of the participants were already known to the researchers from stakeholder engagement meetings. Twenty-four potential participants were initially approached and twelve

participants were recruited. In accordance with the principles of reflexive thematic analysis (TA), recruitment was ended when the targeted sectors had been represented and when high richness and relevance of data had been achieved (rather than aiming to achieve full data "saturation") [33]. One potential participant explicitly declined to participate, citing shortage of time.

### 4.3  Data collection

Semi-structured individual interviews of approximately 45 minutes (ranging from 30 to 60 minutes) were conducted online (on Zoom) in Spanish by one researcher (JF). Audio was recorded and transcribed automatically, then the transcripts were checked, corrected, anonymised and translated to English. The topic guide consisted mainly of open-ended questions designed to describe the participant's work and elicit their experiences of barriers to accessing care and unmet needs in mental health. Additional prompts were derived from topics identified in the literature review and in conversations with gatekeepers: these included migration status and health provider affiliation, understanding of the right to healthcare, discrimination, and migrant attitudes to seeking help. The use of psychoactive substances was added as a prompt topic after this was repeatedly mentioned in early participant interviews. The interview topic guide is added in S1 File.

### 4.4  Data analysis

We used reflexive TA, as described by Braun and Clarke [32,33], as an appropriate method for understanding people's experiences, understanding and representation of phenomena. Transcripts and reflective notes were organised using the NVivo12 software programme. We followed six phases of analysis: familiarisation with the data, generation of initial codes, searching for themes, reviewing themes, defining and naming themes, and production of the final report. Coding was performed line-by-line and following an abductive process: some codes were pre-determined from the topic guide and on the basis of previous research findings, while others were generated openly from the data, permitting new and unexpected insights [34]. A second coding stage followed to clarify code definitions, amalgamate duplicates, and divide large codes. Codes were compared and grouped to generate themes, defined as "patterns of shared meaning, united by a central concept or idea" rather than "topic summaries" [33]. Coding and initial theme development was performed by JF; codes were reviewed by and discussed with CG; final theme development was discussed between JF and CG. We did not seek formally to validate our findings through techniques such as triangulation or participant checking, which may be regarded as methodologically "incoherent" with reflexive TA [32]. However, we did review our findings in the context of existing evidence, as described in section 6.

### 4.5  Reflexivity

JF is a male British researcher whose professional background is as a primary care doctor (General Practitioner) in the UK, with additional expertise in mental health and substance misuse and an interest in improving healthcare access for marginalised populations. He speaks Spanish fluently as a second language. He has Master's-level training in qualitative research methods. JBM is a female Colombian researcher who studied Biology and has a Master's degree in Clinical Epidemiology. She is an assistant professor at the Faculty of Medicine of the Cooperative University of Colombia, Pasto campus, where she leads research projects focused on health of vulnerable populations. CG is a female British public health lecturer at the University of Manchester with experience in conducting and teaching qualitative research. We note the potential influence of the research primarily being performed by a cultural outsider, which may present a barrier to access and frank conversation, but may also have a positive effect if the researcher is regarded as independent from local healthcare systems [35]. There is a risk that non-native Spanish speakers may make errors of misunderstanding or miss nuances [36]; this was mitigated by the availability of a native speaker (JBM) to assist with any uncertainties. JF kept a reflective diary to provide an auditable account of influences and decisions at all stages of research, and reflective notes were included in data analysis.

In reflexive TA, the researcher is encouraged to consider their own role in shaping the generation of knowledge [32]. In conducting the interviews, JF noted the sense of urgency and emotional intensity conveyed by the participants, and felt the pressure of their hopes that the research might increase awareness of their predicament. This has influenced the generation of themes which may appear a call for action, both in terms of resource allocation and further research.

## 5. Results

### 5.1 Description of participants

Table 1 characterises participants by type of organisation and principal type of migrant population with whom they work, to aid understanding of how issues may affect the population in residence or in transit differently. All participants worked in the cities of Pasto or Ipiales. One was employed by a government agency in a role focused on helping migrants to navigate services. The eleven others worked in the non-governmental sector (five mainly in hostels, six mainly in community outreach programmes, but with some overlapping roles). Further individualised details are withheld to maintain anonymity in a relatively small workforce.

### 5.2 Theme one: The reduction of services for migrants with irregular status

The participants describe their experience of the practical and emotional impact of a funding reduction in services for migrants with irregular status in Nariño. This reduction occurred during the period of the study, as a result of the change in migratory flow within Colombia. Specific examples were the closure of shelters, food provision sites and hostels, including facilities where the participants themselves had worked.

> Because the big agencies are leaving and a very large population is left, under-served in terms of health and in terms of mental health… Migration has gone down, it has decreased, but it has not disappeared and it has not decreased on a large scale. So, it's a call that I want to make, that you turn your eyes here, the southern border is a difficult border. (A-2)

Several interviews convey a sense of urgency and frustration in the face of the withdrawal of services for the irregular population, even expressing emotions of desperation and abandonment.

**Table 1. Participant characteristics.**

| Participant | Type of organisation | Principal type of migrant population |
|---|---|---|
| A-1 | Governmental | Resident and transit |
| A-2 | Local, Venezuelan-led | Resident and transit |
| A-3 | Local, Venezuelan-led | Resident and transit |
| A-4 | Local, Venezuelan-led | Resident and transit |
| A-5 | Local | Transit |
| A-6 | Local | Transit |
| A-7 | Local | Transit |
| A-8 | Local | Transit |
| A-9 | National | Resident and transit |
| A-10 | International | Resident and transit |
| A-11 | International | Transit |
| A-12 | International | Resident |

Some participants note a change in official attitudes to migrants in transit, identified in a change of terminology:

> from the language that is now used, right?, of stating, telling them that it is no longer a migration, but a culture of theirs of walking… So, since they are supposedly "used to walking", then there is no longer access, there is no more aid, there is no longer any kind of service. (A-7)

This theme is crucial because all of the participants state that humanitarian cooperation agencies – rather than the Colombian state – have been the main providers of healthcare for irregular migrants in transit:

> Because everything has been left under the response of the cooperation agencies. Without cooperation there would not have been an answer for this population. (A-5)

Many examples of the work of humanitarian agencies are discussed, including psychosocial workers visiting hostels, escalation to care by psychologists, support on the transit route through Nariño and training workshops. Fears are expressed as to how it will be possible to access care if the agencies continue to withdraw and leave altogether:

> if all the social organisations in the city leave, for me it's going to be chaos or a disaster. (A-11)

A particular concern is the absence of services for unaccompanied men in transit:

> We stopped serving unaccompanied men and we have only dedicated or specialised in serving family units. The issue is that we don't have the resources at this time to respond. In the whole department of Nariño or at the national level, there are no resources to serve this population. (A-8)

Several participants describe distress at being unable to provide care to unaccompanied men, due to the specific constraints of their funders. They communicate the ethical challenge of having to exclude a group they recognise as being in need. The participants express differing opinions as to which group of migrants has the most pressing mental health needs, with women, children and LGBT people also identified as highly vulnerable groups. In general, and understandably, participants stress the needs of the particular group with whom they work. This study does not seek to evaluate the relative vulnerability of different sectors of migrants in transit, and it is clear that the whole population has high levels of need. However, these interviews suggest that unaccompanied men are notably lacking in access to care.

### 5.3 Theme two: The vicious cycle of psychoactive substance use

A dominant, recurrent preoccupation of the participants is the use of psychoactive substances by some migrants and its relation to mental health. In several interviews, when asked about "mental health", participants immediately talked about substance use before mentioning other mental health symptoms. Substance use is reported particularly among Venezuelan men with irregular status who are not accompanied by a family unit and who are in transit on foot. Some participants claim that in this specific group, substance use is very common. Drugs mentioned include marijuana, cocaine (and its derivative basuco), an inhaled solvent (bóxer) and amphetamines. For this vulnerable population, there is essentially no available treatment for addiction, other than the emergency department in cases of overdose.

Substance use is viewed as a health concern in its own right, through the effects of intoxication and addiction. Moreover, it is a problem which is linked to and exacerbates other challenges: it may begin as a response to suffering, but then causes additional suffering to the user and others, then excludes them further from the possibility of support.

One participant wished to describe an encounter that had deeply affected them emotionally, with a 17-year-old boy in transit:

> (H)e told me that he didn't use to be an active drug user, but that he started using when he decided to leave Venezuela as a caminante (walker). Having so much time to get to the point where we were, he spent more than a month and a half walking, and he said that he had already run out of resources, he no longer had enough to eat and that it was much easier for someone to be able to buy, to access drugs than to be able to buy a plate of food. (A-4)

Participants state their impression that in most cases the men started using drugs after leaving Venezuela, while in transit, as a means of tolerating physical hardship such as hunger, cold and exhaustion. Moreover, substances are used to provide relief from emotional suffering: migratory grief, anxiety, uncertainty and experiences of hostility. Drug use is seen as self-medication, a response to unmet needs, a way to abandon oneself and to endure. For this man, while substance use was adopted as a means of dealing with distress, it ultimately removed him from potential support systems:

> (he) decided not to communicate with his family anymore while he was in transit because he didn't want them to see him as he was affected. (A-4)

For the participants, the use of substances by people in their care causes specific difficulties in how they perform their work. Users are described as restless, having lost their ability to reason or make decisions, placing them at further risk of harm, including reported cases of suicide while intoxicated. Some interview participants report feeling at personal risk of violence and aggression in the workplace:

> Sometimes someone gets very aggressive and also someone gets in a state, they can't be in the hostel. You have to go out, you must be on the street, walking, walking around a lot. That is, that person is not stable when using this type of substance. (A-11)

The participants who have worked in transit hostels, where substances are not generally permitted, convey frustration and moral distress at how addiction places migrants literally, physically outside of a potential place of safety and psychological support:

> Now, imagine the situation where they prefer to sleep on the street, but they are going to consume; to stop being in one place, a protective space where they are going to receive food, where they can sleep with a blanket or a roof, but the only circumstance was that they could not consume. (A-7)

The harmful effects of psychoactive substances are wider than on the individual users, and therefore also affect participants working in settings other than transit hostels. In the resident migrant population, substance use by male partners is identified as a major factor in high rates of violence against women:

> So, every time we enquire about violence, the consumption of psychological substances comes out. (A-12)

It is implicated in harming the mental health of children who may be exposed directly to drug use and who witness the effects of family breakdown and violence. In a wider sense, it contributes to popular xenophobic stereotypes of Venezuelans as drug-addicts and criminals: this, in turn, worsens stigmatisation and discrimination which can further harm mental health.

### 5.4 Theme three: Competing priorities prevent migrants from attending to their mental health

Many of the participants work in roles which support migrants to identify their health needs and to navigate state health services. They describe the common challenge that arises when economic necessity outcompetes tending to health

needs. For migrants in residence who have achieved affiliation with the healthcare system, attending appointments means time out of paid employment, and potentially costs for follow-up care. Financial imperatives impose choices in which attending to one's mental health is likely to lose out:

> That is to say, their time is destined to work in jobs that surely pay them well below what they legally should pay in Colombia and that does not allow them to have time either. So, they either go to a psychologist or eat, or they go to a psychologist or pay their rent, or they go to a psychologist or they feed and clothe their children. (A-9)

Migrants in transit are even more focused on obtaining what is essential for survival:

> So, the priority for them is to feed themselves and have shelter. None of them think about their mental health, no, not in my experience. (A-8)

For participants working in transit hostels, a major obstacle is the sense of urgency felt in the migrants' impulse to continue the journey, even at the cost of their health.

> They're just worried about getting there, getting there fast, yes. In other words, no matter where they go, they want to get there fast… we could say, they're thinking about their mental health as well because, well, safety is integral, right? But, no, what they think about is just getting there fast. (A-6)

Not attending to one's mental health appears less a choice than an essential consequence of being in transit.

Despite the immense challenges, some participants describe powerful moments of professional satisfaction when they are able to provide non-specialist psychological support:

> "I'm leaving here," he said, they used to say a lot, "I'm leaving here happy because they not only gave me a night's lodging, but because they listened to me". (A-7)

There is a moving description of a workshop held in a transit hostel:

> There have even been scenarios where they have been able to talk about the pain that migration has meant for them, even in tears. There has been catharsis and it has been like a group meeting where the single men have been able to have dialogue about the challenges and risks that they have had to live… (A-5)

These accounts provide insights into how the community workers can enable spaces, however briefly, for expression of emotions. They also challenge the assumption that the migrants in transit are not interested in attending to their mental health, if they should have the opportunity to do so.

### 5.5 Theme four: Rejection in healthcare institutions

Participants highlight that Venezuelan migrants may find themselves rejected in healthcare institutions by being denied the care to which they are legally entitled. Their reports combine direct observation when accompanying migrants to healthcare institutions and narrated accounts told to them by individual migrants. While second-hand accounts in particular must be interpreted with caution, the interviews provide important insights into how institutional barriers are perceived by those in frontline community roles.

Although regularised migrants who hold the TPP should be able to access healthcare in the same way as Colombians, participants repeatedly state that this care may be denied. Two participants describe cases of the entrance to healthcare

centres being physically blocked to Venezuelans by non-clinical staff. Irregular migrants, without the TPP, are only able to access healthcare via the emergency department, but there, too, they may be denied care:

> Can they go to the emergency department? Yes. Can they manage to be taken care of? Not always. (A-9)

One hostel coordinator describes overcoming this barrier by sending migrants preferentially to one centre where they are likely to receive care rather than to another where it is likely to be denied.

The participants attribute this primarily to lack of knowledge on the part of staff (clinical and non-clinical) in healthcare institutions. Several participants state that many employees do not understand the implications of the TPP or the right of all migrants to emergency care, and therefore migrants may be sent away without having received medical attention. The one participant who is a government employee provides insight into how policy and staffing changes may interact to exacerbate this situation:

> …today they are there for a period of time and when you have already developed a process with these people of awareness, training and even, well, of disseminating the rules and how things work, they change it. And you have to start a process all over again, right? (A-1)

However, as well as lack of knowledge, the issue of discrimination within healthcare institutions is raised as a concern:

> Sometimes one feels that there is a certain discrimination, yes, in, maybe, maybe, maybe… I feel that the officials, some of them, are unaware of all the regulations related to the care of the migrant population. But there is some discrimination in the sense that they prefer, in some cases, to provide support to the Colombian population and leave the migrant population behind. (A-10)

There is a hesitancy in making this accusation, which was noted during the interview and is evident verbally in the repeated "maybe". This was a topic which appeared uncomfortable for several participants, and no clear consensus emerges on the existence of discrimination. Some state directly that xenophobia within health institutions exists and is a barrier to equitable care:

> Uff, xenophobia. In other words, when you are Venezuelan, well, they send you to wait much longer. (A-3)

Some note that the subject is outside of their personal experience, so they can't make a judgement. One Venezuelan community leader argues that discrimination against migrants in healthcare has historically been a major problem but that it has now reduced significantly.

One participant describes her experience, in supporting female migrants, of how even isolated incidents of discrimination may have wider effects in deterring the migrant community from seeking to attend institutions:

> I have identified that among the migrant population there are very quick channels for the dissemination of information… But this means that … this criterion is generalised to the whole hospital, and that is why women, in many cases, say, no, I really prefer not to go before they discriminate against me, before they stigmatise me. (A-12)

This example links this theme to the wider subject of xenophobic rejection experienced by Venezuelan migrants – in everyday life, outside of the healthcare context - which resonates throughout the interviews as a major factor causing harm to their mental health.

## 6. Discussion

### 6.1 Discussion of theme one

A major strength of our study is that it provides insights into how community workers experience the constraints and uncertainties associated with shifting policies and funding for humanitarian care for migrants. Irregular migratory status is identified globally as a barrier to mental healthcare access [12]. For Venezuelans in Colombia, irregular status is one of the most significant barriers to mental health care, as irregular migrants are only able to access state healthcare at the emergency department or in services provided by humanitarian organisations. While Colombia has provided enhanced opportunity for regularisation through the TPSV, it is crucial to remember that many migrants have not been eligible for this process, including those arriving after May 2023 (see section 2) and the large number arriving through illegal entry points (see section 3). The reduction of services provided by the non-governmental sector, described by our study participants, is likely to intensify the barrier to access and the extent to which the mental health needs of irregular migrants remain unmet. Since our research was performed, this has become an even more pressing concern due to the decision of the USA Trump administration to reduce foreign aid funding, with severe consequences for NGOs working with migrants in Colombia [37].

The claim of the participants that mental health care for irregular migrants in Nariño is entirely provided by humanitarian and community organisations – rather than the state – is supported by a recent report on healthcare access for irregular migrants across Colombia [6]. That report also highlights potential problems of this strategy including its limited geographical coverage and uncertainty of how services will be provided if the agencies leave. Clearly, funding for humanitarian programmes in Colombia may be redistributed as a response to changes in migratory flow, and our study does not seek to evaluate the relative funding needs of different regions of Colombia. However, the concerns voiced about the lack of mental healthcare for irregular migrants in Nariño should humanitarian organisations continue to withdraw are striking.

Our study participants identify irregular unaccompanied men in transit in Nariño as a group without access to the healthcare provided via hostels. Being unable to provide much needed care to this group was a clear source of distress for several participants. This is supported by reports from other regions of Colombia that donor aid is often specifically allocated for projects for mothers and children, leaving other groups potentially unserved [6]. This is particularly concerning given our findings of harm associated with psychoactive substance use in unaccompanied men. Male migrants in other settings have been noted to have specific health vulnerabilities [38] while paradoxically being regarded as "less worthy" of receiving help [39]. There is an urgent need for further research to evaluate the specific mental health needs of Venezuelan men in transit in Colombia.

### 6.2 Discussion of theme two

Substance use in displaced populations has been insufficiently studied globally: a 2016 systemic review noted that research has been disproportionately conducted in high-income countries rather than LMICs and has focused predominantly on alcohol [40]. A 2024 narrative review found knowledge gaps related to links between displacement, substance use prevalence and varying patterns of risk and resilience across different cultures [41].

In the USA, adolescent Venezuelan immigrants have been identified as at high risk of substance use as measured by intentions and normative beliefs, with a suggestion that substance use is associated with "cultural stress" caused by a negative reception and discrimination in the destination country [42]. In Colombia, a survey of professionals working with Venezuelan migrants identified psychoactive substance use as a common problem [43]. In Venezuelan immigrant sex workers in Colombia, psychoactive substance use has been identified as a strategy of coping with stress and anxiety encountered during sex work [44]. Our study adds to research on this topic by emphasising the extent of psychoactive substance reported in the population of unaccompanied male migrants in transit, and by exploring how substance use interacts with other challenges of the migratory experience. Substance use is described as a response to psychological

distress, leading to disorders of addiction and intoxication for which there is no adequate service for irregular migrants, and serving as a barrier which prevents migrants from accessing other services which might benefit their mental health.

We note that caution is required in interpreting our findings. Our study reports individual community worker perceptions of substance misuse in this population, and particularly its emotional effects on the workers related to perception of risk and frustration at being unable to help. We do not provide the direct accounts of the migrants themselves nor quantitative evidence of the scale of the problem. Our findings should not be generalised to the migrant population and should not be used to feed discriminatory stereotypes about Venezuelan men as drug users. Rather, we suggest there is an urgent need for further research on the scale and potential treatment of substance misuse in migrants in transit.

### 6.3  Discussion of theme three

Socioeconomic structural and logistic factors have been shown to reduce mental health help-seeking amongst refugees internationally and in diverse income settings [38]. For Venezuelans in Colombia, the ability to access healthcare in general is impaired by insecure employment [7,24], unstable or inadequate accommodation [14,24], poor access to transport [5] and socioeconomic vulnerability in general [8,45,46]. Despite the efforts made by the Colombian state to integrate Venezuelans into the health and welfare system, as described in section 2, our study demonstrates that serious socioeconomic barriers remain for Venezuelan migrants in Nariño.

This study provides insight into the difficulties faced by community workers in Nariño in trying to facilitate access to mental health care for migrants in transit, identifying the condition of transit itself as a fundamental barrier. It is recognised at the international level that most mental health support is developed in contexts which differ from the reality of refugee life, and that services delivered for transit populations usually lack continuity of care [47]. Psychological therapy for mental health problems generally involves a prolonged course of treatment, and even briefer programmes "scaled" for adapted use in LMICs tend to involve 5–6 sessions [48]. Our study contributes evidence of the impracticality of providing standard evidence-based psychological care to migrants whose main priority is rapid transit.

The question of how to provide mental health support in the transit context remains an active subject of research with a lack of expert consensus [47]. Our participants identify informal psychological interventions in the setting of a transit hostel – such as providing migrants an opportunity to share their experiences – as a potential facilitator of care. While our study does not seek to determine the effectiveness of any specific intervention in transit hostels, it provides a real-world insight into what can be achieved despite logistical challenges.

### 6.4  Discussion of theme four

An important limitation of our study is that we did not interview employees in healthcare institutions, therefore our conclusions about discrimination or lack of knowledge within those institutions must be drawn tentatively and considered in relation to existing evidence. What has emerged, through reflexive TA, is an understanding of how these barriers are perceived by frontline community workers.

Lack of knowledge by healthcare institution staff of migrants' legal rights is reported as a barrier to health care elsewhere in the literature [6,10]. The specific example described by two participants from our study – of the front door of a healthcare facility being barred to Venezuelans by non-clinical staff - has also been reported in recent research from other Colombian cities [6]. Our study contributes further evidence suggesting an important deficiency in training within Colombian health institutions which risks impeding access for migrants. This may reflect failures at systemic level, rather than on the part of the individual employee.

Significant levels of xenophobia towards Venezuelan migrants have been reported in various South American countries including Colombia, Peru and Ecuador [49]. Discrimination experienced specifically in healthcare settings has been noted as a barrier to mental health care for migrants globally [12] and has been repeatedly reported in Colombia [6,8,10,23]. However, contradictory findings exist: a recent study with community workers supporting migrants in a rural region of

Colombia noted that none of their participants reported that xenophobia affected the quality of healthcare [5]. It is important to emphasise that access to mental health services can be very difficult for Colombian citizens themselves [19] and that fear of the collapse of local services has been reported as underlying some discriminatory attitudes to migrants [49]. A strength of our study is that it captures the complexity of the phenomenon of discrimination, conveying a range of opinions on this issue including the suggestion of a recent reduction in the problem. We do not claim to prove to what extent such discrimination may be occurring, but this study contributes to the impression of an ongoing problem which requires attention at the national level.

## 7. Limitations

An important limitation of our study is that the participants predominantly represented NGOs and the humanitarian sector, with only one government employee included. This was not the intention of our participant selection criteria, but rather a result of the fact that most potential participants who met the inclusion criteria did not work in the state sector. Additionally, we chose not to interview people who worked in healthcare institutions, as outside of the remit of this study. This introduces the risk of potential bias against the Colombian state health system, which is important when considering the controversial topics of institutional knowledge gaps and discrimination. We mitigated this risk by attempting to explore divergent and nuanced views on this topic. While our study provides rich data from an important sector of the workforce, the inclusion of other perspectives would enable a broader understanding of the phenomena described. Further research on this subject could usefully involve interviews with staff in healthcare institutions and strategic leaders in state agencies.

The use of remote interviews permitted flexibility in scheduling, but introduced potential limitations. Firstly, while all interviews were completed as planned, in some cases the internet connection quality was variable causing delays and interruptions which may have impaired the frankness of the conversation. Secondly, during recruitment, it was noted that some potential participants expressed apprehensiveness about internet connectivity which risked under-representation of less well-funded organisations with fewer technological resources. This risk was mitigated by stopping data collection only when the target sectors of the workforce had been included.

## 8. Conclusion

To the best of our knowledge, this is the first qualitative interview-based study in Nariño of the experiences of community workers in supporting Venezuelan migrants to access care for mental health problems. Our use of reflexive TA provides important insights into the lived experiences of community workers in this region of key interest on Colombia's southern border, particularly as they confront a reduction in the provision of migrant health services. We describe striking findings related to the pervasive problem of psychoactive substance use by unaccompanied male migrants in transit and of the extent to which the mental health needs of unaccompanied migrant men in transit are unmet: these are topics which require further research urgently. The results contribute to the wider body of evidence on barriers experienced within healthcare institutions and factors which compete with healthcare as a priority for migrants. We suggest there is an ongoing need for health provider institutions to address the training needs of staff in the legal entitlement of Venezuelan migrants to healthcare and to challenge any discriminatory attitudes.

## Supporting information

**S1 File. Interview topic guide.**
(DOCX)

## Author contributions

**Conceptualization:** John Fitton, Julie Benavides-Melo, Christine Greenhalgh.

**Formal analysis:** John Fitton, Christine Greenhalgh.

**Investigation:** John Fitton, Julie Benavides-Melo.

**Methodology:** John Fitton, Christine Greenhalgh.

**Supervision:** Julie Benavides-Melo, Christine Greenhalgh.

**Writing – original draft:** John Fitton.

**Writing – review & editing:** Julie Benavides-Melo, Christine Greenhalgh.

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
