## [Decision Letter · Decision Letter 0]

15 Jan 2026

PMEN-D-25-00474

Barriers to access and unmet needs in mental health care for Venezuelan migrants in a southern border region of Colombia: the experiences of community workers.

PLOS Mental Health

Dear Dr. Fitton,

Thank you for submitting your manuscript to PLOS Mental Health and I am very sorry for the severe delay in sending a decision to you. This was due to difficulties securing reviewers and then delays over the festive season. After careful consideration of the reviewer reports, we feel that your paper has merit but does not fully meet PLOS Mental Health’s publication criteria as it currently stands. Therefore, we invite you to submit a revised version of the manuscript that addresses the points raised during the review process.

Please address all of the comments that reviewers have raised, which you can find at the end of this email and attached.

We look forward to receiving your revised manuscript.

Kind regards,

Karli Montague-Cardoso

Staff Editor

PLOS Mental Health

Journal Requirements:

1. Please provide additional information regarding the considerations  made for the prisoners included in this study. For instance, please discuss whether participants were able to opt out of the study and whether individuals who did not participate receive the same treatment offered to participants.

2. Please ensure that your Ethics Statement is available in its entirety at the beginning of your Methods section, under a subheading 'Ethics Statement'. It must include:

i) The name(s) of the Institutional Review Board(s) or Ethics Committee(s)

ii) The approval number(s), or a statement that approval was granted by the named board(s)

iii) (for human participants or donors) - A statement that formal consent was obtained (must state whether verbal/written) OR the reason consent was not obtained (e.g., anonymity).

Additional Editor Comments (if provided):

Reviewers' comments:

Reviewer's Responses to Questions

**Comments to the Author**

1. Does this manuscript meet PLOS Mental Health’s publication criteria? Is the manuscript technically sound, and do the data support the conclusions? The manuscript must describe methodologically and ethically rigorous research with conclusions that are appropriately drawn based on the data presented.? Is the manuscript technically sound, and do the data support the conclusions? The manuscript must describe methodologically and ethically rigorous research with conclusions that are appropriately drawn based on the data presented.

Reviewer #1: Partly

Reviewer #2: Yes

2. Has the statistical analysis been performed appropriately and rigorously?

Reviewer #1: N/A

Reviewer #2: N/A

3. Have the authors made all data underlying the findings in their manuscript fully available (please refer to the Data Availability Statement at the start of the manuscript PDF file)?

The PLOS Data policy requires authors to make all data underlying the findings described in their manuscript fully available without restriction, with rare exception. The data should be provided as part of the manuscript or its supporting information, or deposited to a public repository. For example, in addition to summary statistics, the data points behind means, medians and variance measures should be available. If there are restrictions on publicly sharing data—e.g. participant privacy or use of data from a third party—those must be specified.requires authors to make all data underlying the findings described in their manuscript fully available without restriction, with rare exception. The data should be provided as part of the manuscript or its supporting information, or deposited to a public repository. For example, in addition to summary statistics, the data points behind means, medians and variance measures should be available. If there are restrictions on publicly sharing data—e.g. participant privacy or use of data from a third party—those must be specified.

Reviewer #1: Yes

Reviewer #2: Yes

4. Is the manuscript presented in an intelligible fashion and written in standard English?

Reviewer #1: Yes

Reviewer #2: Yes

Reviewer #1: The manuscript addresses a highly relevant issue. The barriers to mental health care for Venezuelan migrants, the localized focus on Nariño as a transit context, and the decision to explore these challenges from the perspective of community workers, constitute clear strengths. The study contributes valuable qualitative insights into assess the gaps regarding MH services.

Overall, the research is ethical, methodologically appropriate, and results are clearly reported in a systematic manner easy to follow. With revision, the manuscript has strong potential to make an important contribution to the literature on migrant mental health. Here some recommendations:

1. Clarify and strengthen the conceptualization of what is defined as “access to mental health care”. Throughout the manuscript, access is discussed on various terms: legal, institutional, service availability/usage/continuity. I believe that defining "access" can improve analytic precision and specially avoid overgeneralization, (that to my point of view is the major flow I find in the manuscript). I recommend explicitly defining what is meant by “access” early in the manuscript to set the tone for results and interpretations.

2. Regarding as well overgeneralization, the identification of psychoactive substance use among unaccompanied men in transit is one of the most important and original contributions of the study. However, results suggests generalization of substance use in this group, which is risky to asseverate specially having into account that the information collected comes from third-party perceptions. I recommend clearly emphasize that these findings reflect community workers’ experiences, and give it value as a very relevant entry point for further research needed.

3. Acknowledge and contextualize the imbalance in participant representation with predominance of NGO and humanitarian-sector participants, with minimal representation from state health institutions. While the limitation is acknowledged, the conclusions would benefit from clearer distinctions between observed practices and perceived barriers, and from a more cautious framing of claims about the state health system, again related with overgeneralization.

4. Minor stylistic and grammatical edits are recommended, particularly to shorten long sentences. Also, throughout the manuscript, the term “ignorance” is repeatedly used to describe provider-level barriers. I recommend replacing this term with more precise and neutral language (e.g., limited awareness, knowledge gaps, lack of knowledge or insufficient training), which are more constructive and focus on systemic or structural factors rather than individual failings.

Reviewer #2: This manuscript reports a qualitative study based on 12 interviews with community workers in Colombia, exploring barriers to mental health care access for Venezuelan refugees. The topic is important and timely, and the paper offers useful insights from a frontline, practice-based perspective. Congratulations on your work, and thank you for inviting me to review this work. The authors have provided a well written study, the work is thoroughly performed and keeps a solid, logic disposition that makes it easy for the reader to follow.

Below find my suggestions for revisions that need to be done to be a valuable contribution to this research field

Major comment: Participants and key concept (“community workers”) need a clear definition and stronger justification

A central issue is that the paper does not sufficiently define who “community workers” are in the Colombian health system, what their role/mandate is, and why they are the appropriate participants for this study. While the background provides a strong description of Venezuelan refugees and the barriers they face, the reader is left unsure about the study population and the meaning of the key term that structures the entire analysis. Another main concern is, why is the community workers the subject of the study and not refugees themselves, specially on topics that can be potentially stigmatizing – topics are being discussed two steps apart from the affected population (rseachers are discussiong refugees lived experience through community workers) – this question is elaborated further under point 4 – hopefullt it can help in arguing choice of population

Please revise (preferable in setting under methods):

1) Define “community workers” early and precisely (setting/background, not only later)

Because the participants are community workers (and no other group), the concept needs to be introduced clearly in the background/setting. At present, the refugees are well defined, but the participants are not.

Please clarify:

• Are community workers government employees, NGO staff, CHWs/lay workers, outreach workers, social workers, or a mixed group? Can it also be Venezuelan community leaders (as implied on line 251)?

• What are their core responsibilities (screening, referral, psychoeducation, case management, outreach, navigation, training dissemination, etc.)?

• What is their position in the system (formal health system vs humanitarian sector; paid vs volunteer; clinical vs non-clinical; scope of practice)?

• This is particularly important because “community worker” can mean very different things across LMIC contexts. Without an explicit local definition, the reader may import assumptions from other settings and misinterpret the findings.

Example: later quotes suggest heterogeneity in mandates (e.g., a government employee discussing training dissemination). A short description of role categories would make those quotations and implications much clearer.

2) Align the methodology claims (“phenomenology” / “lived experience”) with what is actually studied

The manuscript describes a phenomenological aim to understand the world as “experienced by people” and participants’ “lived experience”. However, much of the findings are community workers’ descriptions of refugees’ experiences (i.e., second-hand accounts), while the stated aim emphasises community workers’ challenges and ethical difficulties.

It would help to clarify:

• Is the analytic focus the community workers’ own lived experience of providing support (e.g., moral distress, role conflict, safety, resource constraints)?

• Or is the focus primarily on their perspectives on refugees’ barriers and needs?

• Right now, there is some tension between the methodological framing and what is presented as results, i.e. the aim claims to explore community workers challenges with barriers Venezuelan face in accessing mental health care – but it is not how th results is presented.

3) Strengthen inclusion/exclusion criteria and provide a rationale for excluding those with mental health education. The inclusion/exclusion criteria need to be more explicit and better justified, especially the exclusion of people with mental health education/training.

As written, this exclusion raises questions because in many LMIC and humanitarian contexts, community workers (broadly defined) are a core part of the mental health and psychosocial support ecosystem, and some may have structured training or hybrid roles.

Excluding people with mental health education could remove precisely the group most involved in MHPSS delivery—unless the aim is specifically to capture non-specialist perspectives.

Please clarify:

• What counts as “mental health education” (formal degree? short course? WHO mhGAP training? in-service training?)

• Why is this exclusion necessary for the research question?

• How does excluding this group affect the transferability of the findings to the broader “LMIC context” that the authors repeatedly refer to?

4) Explain why community workers are the study subjects (and not refugees themselves)

The paper would benefit from a clearer statement of why community workers are the primary participants rather than refugees. If the authors’ contribution is meant to be a “narrative” of the refugee situation mediated through workers, that should be explicitly framed as both a strength and a limitation. If the reason is that community workers are easier to recruit than refugees, please elaborate on why community workers are an important source anyway.

In particular:

• What unique insight do community workers provide (health caresystem navigation, implementation barriers, ethical dilemmas, policy changes, service fragmentation)?

• What does this perspective allow the study to contribute beyond what is already known from refugee-centred literature?

5) Clarify novelty: Themes 1–2 read as confirmatory; Themes 3–4 feel more original

Relatedly, the introduction/background already cover major barriers, so the first two themes may feel largely confirmatory. The more novel contribution appears to come from the later themes on substance use and what that implies for the Venezuelan refugees and around system change in Columbia which influence the refugees migration status, such as: irregularity/uncertainty in status, and how shifting policies increase vulnerability and constrain support. Moreover, what does the substance use mean for the community workers, their moral dilemmas and so on? Additionnaly, the huge etchial dilemma that the services seem to focus on families and not on unaccompined young men – who in many migrant contexts are the largest population

The manuscript may be stronger if it foregrounds and develop theme 3 and 4 more explicitly as the key added value, and don’t put “equal weight” (as it is now) on all four themes in result and discussion.

6) Interview guide is missing as supplementary material

The other major revision is on the reflexive thematic method

My recommendation is that the authors reflect on whether to change method for data analysis and justify change on analysis, or to re-do analysis according to RTARG, below are my arguments for why this suggested change:

Braun and Clarke have recently introduced the Reflexive Thematic Analysis Reporting Guidelines (RTARG) and argue that checklist-driven reporting (e.g., COREQ) and “validation” procedures can pull reflexive TA towards a (post)positivist logic that is not coherent with the method. In other words, reflexive TA does not aim to verify themes as if they were objective findings; it emphasises the researchers’ interpretative role and subjectivity as part of the analytic process itself.

Lines 180–181 state: “The validity of the study findings was evaluated by triangulation with existing local and national evidence.” This is currently unclear and would benefit from a more explicit explanation of what “triangulation” means here. For example: what exactly was triangulated (themes, interpretations, descriptive claims)? Which sources were used? And was the aim to situate findings in policy/evidence or confirmation/validation (checking whether findings are “true”)?

As written, the phrasing reads as if the researchers are attempting to confirm/justify the findings rather than primarily explore the dataset (what comes out from the 12 interviews), which sits somewhat contradict with reflexive TA as described by Braun and Clarke. I also read it as the analysis may be a mix of approaches (which is common): reflexive thematic analysis combined with more descriptive/content-analytic moves, where the goal is closer to manifest description, sorting, and categorising, and where “validity checks” like triangulation are more routinely invoked.

There is nothing inherently wrong with a methodological “mash-up”, but if this is what has been done, it becomes especially important to be explicit and coherent: clearly name the analytic approach (or combination), align terminology with that approach, and justify why each analytic step (including any triangulation/validation claims) was necessary and appropriate within the chosen qualitative paradigm.

For more information on RTARG, which I suggest the authors to use if their intention is to stick to reflexive thematic analysis

Braun, V., & Clarke, V. (2025). Reporting guidelines for qualitative research: a values-based approach. Qualitative Research in Psychology, 22(2), 399–438. https://doi.org/10.1080/14780887.2024.2382244

**Do you want your identity to be public for this peer review?** For information about this choice, including consent withdrawal, please see our Privacy Policy..

Reviewer #1: No

Reviewer #2: No

---

## [Decision Letter · Decision Letter 1]

11 Mar 2026

PMEN-D-25-00474R1

Barriers to access and unmet needs in mental health care for Venezuelan migrants in a southern border region of Colombia: the experiences of community workers.

PLOS Mental Health

Dear Dr. Fitton,

Thank you for submitting your revised manuscript to PLOS Mental Health. After editorial and reviewer assessment, we found the manuscript to be much improved and almost ready for acceptance. One of the reviewers had one final, minor comment that we would like to give you the chance to address. Once you have resubmitted, this will not be sent back to the reviewers again and we will check the minor changes in-house. Thank you for your understanding.

We look forward to receiving your revised manuscript.

Kind regards,

Karli Montague-Cardoso

Staff Editor

PLOS Mental Health

**Journal Requirements:**

**Additional Editor Comments (if provided):**

Reviewers' comments:

Reviewer's Responses to Questions

**Comments to the Author**

Reviewer #1: (No Response)

publication criteria? Is the manuscript technically sound, and do the data support the conclusions? The manuscript must describe methodologically and ethically rigorous research with conclusions that are appropriately drawn based on the data presented.? Is the manuscript technically sound, and do the data support the conclusions? The manuscript must describe methodologically and ethically rigorous research with conclusions that are appropriately drawn based on the data presented.

Reviewer #1: Yes

3. Has the statistical analysis been performed appropriately and rigorously?

Reviewer #1: N/A

4. Have the authors made all data underlying the findings in their manuscript fully available (please refer to the Data Availability Statement at the start of the manuscript PDF file)?

The PLOS Data policy requires authors to make all data underlying the findings described in their manuscript fully available without restriction, with rare exception. The data should be provided as part of the manuscript or its supporting information, or deposited to a public repository. For example, in addition to summary statistics, the data points behind means, medians and variance measures should be available. If there are restrictions on publicly sharing data—e.g. participant privacy or use of data from a third party—those must be specified.requires authors to make all data underlying the findings described in their manuscript fully available without restriction, with rare exception. The data should be provided as part of the manuscript or its supporting information, or deposited to a public repository. For example, in addition to summary statistics, the data points behind means, medians and variance measures should be available. If there are restrictions on publicly sharing data—e.g. participant privacy or use of data from a third party—those must be specified.

Reviewer #1: Yes

5. Is the manuscript presented in an intelligible fashion and written in standard English?

Reviewer #1: Yes

**Reviewer #1:** The revised manuscript has improved following the previous round of peer review. The authors have addressed several of the concerns raised in the initial review, particularly by moderating some claims, improving terminology, and better contextualizing the participant composition. The revised manuscript has improved following the previous round of peer review. The authors have addressed several of the concerns raised in the initial review, particularly by moderating some claims, improving terminology, and better contextualizing the participant composition. The revised manuscript has improved following the previous round of peer review. The authors have addressed several of the concerns raised in the initial review, particularly by moderating some claims, improving terminology, and better contextualizing the participant composition. The revised manuscript has improved following the previous round of peer review. The authors have addressed several of the concerns raised in the initial review, particularly by moderating some claims, improving terminology, and better contextualizing the participant composition.

One aspect that to my point of view, could still be strengthened is the conceptual clarification of “access to mental health care”. Although the manuscript now discusses multiple dimensions of access (e.g., legal eligibility, service availability, utilization, and continuity of care), providing a clearer conceptual definition earlier in the manuscript would improve analytical clarity and help frame the interpretation of the findings, as highlighted in the prior revision.

**Do you want your identity to be public for this peer review?** For information about this choice, including consent withdrawal, please see our Privacy Policy..

Reviewer #1: No

---

## [Editor Report · Decision Letter 2]

24 Mar 2026

Barriers to access and unmet needs in mental health care for Venezuelan migrants in a southern border region of Colombia: the experiences of community workers.

PMEN-D-25-00474R2

Dear Dr Fitton,

We are pleased to inform you that your manuscript 'Barriers to access and unmet needs in mental health care for Venezuelan migrants in a southern border region of Colombia: the experiences of community workers.' has been provisionally accepted for publication in PLOS Mental Health.

Best regards,

Karli Montague-Cardoso

Staff Editor

PLOS Mental Health